# Development of Novel Bioluminescent Biosensors Monitoring the Conformation and Activity of the Merlin Tumour Suppressor

**DOI:** 10.3390/ijms25031527

**Published:** 2024-01-26

**Authors:** Alexander Pipchuk, Tynan Kelly, Madeleine Carew, Christopher Nicol, Xiaolong Yang

**Affiliations:** Department of Pathology and Molecular Medicine, Queen’s University, Kingston, ON K7L 3N6, Canada; alex.pipchuk@queensu.ca (A.P.); 18tnpk@queensu.ca (T.K.); 17mrc7@queensu.ca (M.C.); nicolc@queensu.ca (C.N.)

**Keywords:** Hippo pathway, Merlin, tumour suppressor, biosensor, NanoLuc, NanoBiT

## Abstract

Solid tumours can universally evade contact inhibition of proliferation (CIP), a mechanism halting cell proliferation when cell–cell contact occurs. Merlin, an ERM-like protein, crucially regulates CIP and is frequently deactivated in various cancers, indicating its significance as a tumour suppressor in cancer biology. Despite extensive investigations into Merlin’s role in cancer, its lack of intrinsic catalytic activity and frequent conformation changes have made it notoriously challenging to study. To address this challenge, we harnessed innovative luciferase technologies to create and validate a NanoBiT split-luciferase biosensor system in which Merlin is cloned between two split components (LgBiT and SmBiT) of NanoLuc luciferase. This system enables precise quantification of Merlin’s conformation and activity both in vitro and within living cells. This biosensor significantly enhances the study of Merlin’s molecular functions, serving as a potent tool for exploring its contributions to CIP and tumorigenesis.

## 1. Introduction

Merlin (Moesin, Ezrin, Radixin-like protein) was initially discovered as the protein product of the *NF2* gene in 1993 [1,2]. In line with the two-hit model of tumour suppression, the biallelic mutation of *NF2* leads to the development of neurofibromatosis type II (NF2) [3], a tumourigenic genetic disease characterised by bilateral schwannoma formation along the vestibulocochlear cranial nerve [4]. NF2 patients also display an elevated risk of developing schwannomas at other locations or brain tumours of different varieties (i.e., meningiomas and ependymomas) [5]. Disease penetration is nearly 100%, and most patients experience a shortened overall life expectancy that progresses to hearing loss [3,4]. Currently, NF2 management is primarily focused on surgical intervention. An improved understanding of molecular Merlin function is needed to identify opportunities for targeted therapeutic intervention.

In spontaneous tumours, Merlin inactivation is ubiquitous or near-ubiquitous in schwannomas [5,6] and highly frequent in meningiomas and ependymomas [7,8,9,10]. Genetic *NF2* mutations also occur in roughly half of all mesotheliomas. Merlin loss is also thought to contribute greatly to disease progression [11,12]. Less frequently, Merlin is genetically inactivated in a wide variety of other cancers [13,14,15,16]. In mesotheliomas, breast cancers, and prostate cancers without genetic or transcriptional inactivation of *NF2*, there is evidence of functional Merlin inactivation at the post-translational modification level [14,17,18]. Additionally, Merlin deficiency may contribute to tumour metastasis [19,20,21], drug resistance [22], and sensitivity to an iron-dependent form of cell death known as ferroptosis [23]. Overall, Merlin is an influential tumour suppressor of general importance in cancer biology. However, the cellular signalling events underlying Merlin’s inhibition of tumour progression are not fully understood.

In the context of tumour suppression, Merlin coordinates contact inhibition of proliferation (CIP), the mechanism by which intercellular contacts engage in signalling to stop the proliferative growth of cells. A hallmark feature of solid tumours is the ability to circumvent or overcome CIP [24,25,26]. As such, loss of CIP is regarded as a key event in tumourigenesis. Merlin-deficient cells lose their ability to be contact-inhibited in cell culture and form tumours in vivo, establishing a crucial role for Merlin in tumour suppression [14,21,27]. However, Merlin is a difficult protein to study because of its diverse subcellular localisation, lack of intrinsic catalytic activity, and frequent conformation changes. As such, the molecular functions of Merlin in tumour suppressive cell signalling are not fully understood, despite nearly three decades of research efforts. 

Merlin shares a close sequence and structural similarity to the ERM (Ezrin, Radixin, Moesin) family of actin-cytoskeleton linker proteins [28,29,30,31]. Similar to other ERM proteins, Merlin activity is regulated by a change in conformation. Merlin adopts two distinct conformations: a ‘closed’ conformation characterised by an N-to-C terminal interaction, and a more linear ‘open’ conformation [28,32]. In its closed conformation, the intramolecular Merlin association masks binding sites targeting downstream effectors. As such, this conformation is thought to be inactive in tumour suppression, whereas open-form Merlin is hypothesised to be an active tumour suppressor [8,28,32]. However, there are no crystallographic data for full-length Merlin in either conformation. Speculation regarding Merlin’s conformation–activity relationship has been the subject of intense study and controversy. Unique biochemical techniques must be developed to study Merlin’s conformation–activity relationship more accurately. Merlin’s change in conformation is mediated by phosphorylation at serine-518 (S518). S518 is the most commonly studied site of post-translational Merlin modification and is phosphorylated by P21-activated kinases (PAK) and protein kinase A (PKA) [33,34,35]. Phosphorylation of S518 promotes closed-conformation, inactive Merlin; however, this residue is unlikely to participate directly in the N-to-C terminal binding interface [36]. Instead, phosphorylation is thought to inhibit binding with other upstream regulators of Merlin activity, such as angiomotin, to prevent transition to the open conformation. Due to the well-established influence of phosphorylation at S518 on Merlin activity and its purported conformation, phosphodeficient (S518A; A, alanine) and phosphomimetic (S518D; D, aspartic acid) mutants are commonly applied to study Merlin activity. However, there is currently no method of directly monitoring the impact of phosphorylation on Merlin’s conformational state. 

Furthermore, Merlin is a potent upstream regulator of the Hippo signalling pathway [36,37,38,39,40], an evolutionarily conserved kinase cascade with well-recognised and multifarious roles in protecting against tumourigenesis [41,42,43,44,45]. Merlin potentiates the activity of LATS, the core Hippo kinase [36,46], through a direct interaction coordinating the colocalization of LATS alongside other Hippo components at the plasma membrane [40]. A 7 amino acid, evolutionarily conserved ‘blue-box’ region in Merlin’s F2 subdomain (Figure 1A) is heavily involved in this interaction [36,40,47,48]. Deletion or alanine-substitution of the ‘blue-box’ residues abolishes Merlin’s tumour suppressive capacity [49,50]. Moreover, the ‘blue-box’ residues are masked by the C-terminal tail of Merlin in its closed conformation [36,47], and mutations that lock Merlin into a closed conformation cannot bind LATS and lack tumour suppression [36,46]. Functionally, the Hippo pathway is also well-established as a key downstream signalling mechanism involved in coordinating CIP [51,52,53]. Together, these findings strongly support Merlin’s interaction with LATS as a key mediator of its function. 

We recently developed several bioluminescent biosensors to monitor the levels and activities of the Hippo pathway components (e.g., LATS, YAP/TAZ-TEAD PPI) in vitro and in vivo [54,55,56,57,58,59]. Using these biosensors, we identified many upstream regulators and small molecule (SM) drugs modulating the Hippo pathway for cancer therapies. In this study, we propose using a similar biosensor system as an improved method for studying Merlin activity. In brief, a split luciferase system allows for the development of complementation assays, wherein a functional luciferase is split into two non-functional components [59]. These constituents can be fused onto a pair of interacting proteins, reconstituting a functional luciferase and luminesce upon protein–protein interaction (PPI) between the interaction pair of interest. This system provides a sensitive and accurate method for quantifying a given PPI in real time. Promega recently engineered NanoBiT: a split luciferase system demonstrating improved sensitivity, thermal stability, and size compared to previous split-luciferase technologies [60,61]. NanoBiT is comprised of an 18 kDa LgBiT subunit and a 1.3 kDa, 11-amino acid SmBiT subunit. 

In this work, we developed and validated a split-luciferase system to study Merlin’s intramolecular association. First, we cloned the LgBiT and SmBiT luciferase constituents onto the N and C-terminus, respectively, of full-length Merlin in a unimolecular biosensor system. We hypothesized that, upon N-to-C-terminal interaction and transition to Merlin’s ‘closed’ conformation, this intramolecular biosensor (Mer-Intra-BS) would emit increased luminescent activity, indicating LgBiT and SmBiT complementation (Figure 1C). By contrast, less luminescent activity would be observed when Merlin exists primarily in an open conformation. This system was validated using mutations to disrupt or promote the binding event of interest and, thus, change the biosensors’ expected luminescent activity. This biosensor serves as a powerful tool for studying Merlin’s molecular function in real time and can be applied in future work to offer unique insights into the elusive tumour-suppressive functions of this protein. 

## 2. Results

### 2.1. Development and Validation of Mer-Intra-BS

LgBiT and SmBiT were cloned onto the N- and C-terminus, respectively, of full-length Merlin (Figure 1B). Upon conformation change from an open form to an autoinhibited, closed form, we expect the luciferase constituents of this Mer-Intra-BS to reconstitute a functional luciferase and emit light (Figure 1C). In contrast, we anticipate less luminescent activity of the biosensor system when Merlin exists primarily in an open conformation (Figure 1C). As expected, the Mer-Intra-BS showed a dramatic increase in luminescent activity compared to untransfected (mock) controls, or controls transfected with the pBiT1.1-N vector expressing LgBiT alone (Figure 1D). LgBiT displays a small degree of luminescent activity without having to complement SmBiT. However, compared to LgBiT, about a 149-fold increase in luminescent activity was observed in the Mer-Intra-BS, demonstrating that luciferase complementation occurred as predicted. Expression of the Mer-Intra-BS was also verified by Western blot (Figure 1E).

After confirming luciferase complementation and the successful construction of the Mer-Intra-BS as designed, we validated the hypothesis that luciferase complementation responds to Merlin’s change of conformational state. We first generated a mutant biosensor system that was locked into a closed conformation. Substitution of alanine (A) at position 585 with a tryptophan (W) residue (A585W) promotes interaction between Merlin’s N and C-terminus and stabilises the closed conformation of the protein [36,46]. Therefore, an A585W mutant biosensor system should show increased luminescent activity compared to an identical wild-type (WT) Mer-Intra-BS under the same promoter. After site-directed mutagenesis and cloning of the A585W mutant Mer-Intra-BS, we observed a significant increase in luminescent activity of the A585W biosensor compared to the WT Mer-Intra-BS (Figure 2A). This increase in luminescent activity did not correlate with the increased expression detected by Western blot (Figure 2B). Furthermore, a recent study identified the crucial role of the last two amino acids, glutamic acid (E) and leucine (L), in forming Merlin’s closed form. The Merlin mutant lacking these residues, denoted as ΔEL, adopts an open conformation [46,62]. To explore this phenomenon, we engineered an intramolecular biosensor for the Merlin ΔEL mutant, referred to as Mer-ΔEL-intra-BS. Compared to the wild-type (WT) control, we observed a significant decrease in luciferase activity for Mer-ΔEL-intra-BS (Figure 2C) despite similar expression levels (Figure 2D). These results provide compelling evidence that Mer-Intra-BS accurately reflects a conformational shift within living cells.

After rigorous validation, we established stable cell lines expressing Mer-Intra-BS under the control of a Dox-inducible promoter using lentiviral transgene introduction. Both HEK293T and HCT-116 cells were infected with a lentivirus carrying the Mer-Intra-BS construct. Similar to the results obtained from transient transfection luciferase assays (Figure 2), the induction of Mer-Intra-BS protein expression by Dox led to a remarkable increase in these cells’ luminescent activity (Figure 3), as indicated by the elevated levels of Mer-Intra-BS protein shown in Figure 3. 

### 2.2. Live Cell Bioluminescent Imaging (BLI) Analysis of Mer-Intra-BS Activity

We conducted bioluminescence imaging (BLI) analysis in living cells using cell lines that stably express Mer-Intra-BS. No bioluminescent signals were detected in the absence of Dox (-Dox). In the presence of Dox (+Dox), bioluminescent signals in living cells progressively increased with the rise in cell numbers (Figure 4).

### 2.3. Cell Density Dependent Regulation of Mer-Intra-BS Activity

By employing these firmly established cell lines, our investigation unveiled a dynamic pattern in Mer-Intra-BS signalling. Notably, the signal intensified during cell proliferation at lower densities (1 × 10^4^ cells, 40–50% confluency) in HEK293T cells. Despite maintaining comparable expression levels at different cell densities, after normalization of Mer-Intra-BS expression levels (Western blot analysis) by densitometry analysis using ImageJ, the signal experienced a significant reduction as cell density increased (2–8 × 10^4^ cells, 80–100% confluency) (Figure 5).

This observation suggests that the Mer-Intra-BS signal can monitor Merlin’s contact inhibition effect. The signal is elevated in its closed form at low cell density and is reduced in its open form at high cell density. Furthermore, cell lines maintaining a stable expression of Mer-Intra-BS present promising applications for in vivo xenograft mouse models. The integration of bioluminescent imaging and luciferase analysis in these models offers a robust platform for comprehensive assessments of Merlin’s activity and its influence on tumour suppression.

### 2.4. Characterization of Mer-Intra-BS In Vitro

In-depth characterization of Mer-intra-BS was conducted in vitro and involved the purification of a His-tagged Mer-intra-BS fusion protein in bacterial cells, as illustrated in Figure 6A. Notably, luciferase activity exhibited a positive correlation with the quantities of purified His-Mer-intra-BS used in the assays [correlation coefficient (R) = 0.97; Figure 6B]. This purified biosensor fusion protein is a valuable tool for precisely quantifying and validating proteins, kinases, or drugs that influence Merlin’s intramolecular interaction and activity. Its availability paves the way for comprehensive studies on modulating Merlin’s function and enables a deeper understanding of its regulatory mechanisms and potential therapeutic interventions.

## 3. Discussion

In summary, we developed and validated a luciferase-based biosensor to study Merlin’s conformation changes and activity. Merlin’s conformation–activity relationship has been a subject of controversy that is still unresolved despite intense biochemical investigation. The Mer-Intra-BS provides strong evidence in support of the current Merlin conformation–activity relationship model, wherein the closed-form protein is inactive in tumour suppression [33,34].

The development and validation of Mer-Intra-BS present several significant contributions to our current understanding of Merlin biology. Firstly, the creation of a unimolecular Merlin-NanoBiT biosensor is an exciting proof of concept. This achievement is noteworthy considering that many other proteins, such as the FERM family (e.g., Ezrin and Moesin), also undergo similar N-to-C terminal autoinhibitory conformational changes. Furthermore, the A585W and ΔEL mutants of Mer-intra-BS provide unequivocal evidence, confirming that these mutations, now widely employed, represent closed and open forms, respectively, of Merlin conformations. These findings align with and support results from other studies [36,46]. Thirdly, Merlin activation by transitioning from its closed to open form has been proposed as a mechanism by which Merlin exerts its tumour suppressor function in regulating CIP [14,21,27]. Our findings using Mer-Intra-BS are the first to suggest that Merlin transforms from a closed to an open form during increased cell–cell contact (Figure 5). Lastly, the establishment of two stable cell lines expressing Mer-Intra-BS provides a valuable toolkit for future investigations characterising Merlin’s intramolecular conformation within living cells and xenograft mouse models. These cell lines offer a robust platform for delving deeper into the intricate dynamics of Merlin’s conformational states, facilitating a more comprehensive understanding of its biological functions and opening avenues for potential therapeutic exploration.

There are several promising avenues for exploration utilizing the Merlin biosensor system we developed. Firstly, functional characterisation of cells stably expressing Mer-Intra-BS is essential to confirm its retention of tumour-suppressive activity. Rigorous testing is warranted to establish the biosensor’s efficacy in capturing Merlin’s biological functions. Secondly, the creation of additional mutant biosensor systems and the incorporation of patient-derived substitutions presents an exciting opportunity to investigate the molecular consequences of these variants for Merlin’s intramolecular associations. For instance, certain NF2 patient-derived substitutions are known to impede the phosphorylation of YAP, a process normally facilitated by WT Merlin [36]. Employing point-mutant Mer-intra-BS constructs can directly assess whether these substitutions correlate with decreased luciferase activity, shedding light on the functional implications of these mutations.

Moreover, the Mer-intra-BS system holds the potential for conducting gain-of-function screens, specifically for kinases that regulate Merlin’s intramolecular interaction and tumour suppressor activity. Additionally, the biosensor may be employed in screening endeavours to identify small molecule drugs that disrupt Merlin’s intramolecular interactions and activate its tumour suppressor function, a strategy successfully implemented in our previous work [54,55,56,57]. By delving into these research directions, we can gain valuable insights into Merlin’s intricate regulatory network and explore novel therapeutic interventions for cancer treatment.

## 4. Materials and Methods

### 4.1. Biosensor Design and Construction

Structurally, Merlin consists of an N-terminal FERM domain, a central helical domain, and a C-terminal domain (CTD) (Figure 1A). The Mer-Intra-BS construct consists of an N-terminal LgBiT constituent, a central full-length human Merlin component (accession number NM_000268.3), and a C-terminal SmBiT constituent. Human Merlin was amplified by polymerase chain reaction (PCR). To fuse the SmBiT luciferase constituent onto Merlin, a 33-nucleotide sequence encoding SmBiT was included as an overhang on the reverse PCR primer, alongside a DNA segment encoding a flexible glycine-serine (G/S) region necessary for efficient luciferase complementation. As such, SmBiT and its G/S linker region were incorporated into the C-terminus of the Merlin PCR product (Appendix A). Following PCR, the Mer-SmBiT product was digested and ligated into the EcoR1/Nhe1 sites of pBiT1.1-N vector (Promega, Madison, WI, USA), which contains an N-terminal sequence encoding LgBiT and a G/S linker in frame with the Merlin-SmBiT construct. Thus, a LgBiT-linker-Merlin-linker-SmBiT intramolecular Merlin biosensor was cloned (Figure 1B). This construct was later amplified and cloned into the pcDNA3.1 hygro(+) vector.

### 4.2. Site-Directed Mutagenesis

Site-directed mutagenesis was accomplished by overlapping PCR, as described elsewhere [63]. See Appendix A for a list of the primers used for mutagenesis and cloning.

### 4.3. Cell Culture

HEK293T (human embryonic kidney; ATCC, Cat#CRL-3216) cells were cultured in Dulbecco’s Modified Eagle Medium (DMEM, Cat#D6429; MilliporeSigma Canada, Oakville, ON, Canada) containing 10% Fetal Bovine Serum (FBS) and 1% Penicillin/Streptomycin (P/S; Invitrogen, Burlington, ON, Canada). HCT-116 (human colon cancer; ATCC, Cat#CCL-247) cells were cultured in McCoy’s 5A medium (Sigma M4892), supplemented with 10% Fetal Bovine Serum (FBS), and 1% P/S (Invitrogen). All cells were cultured at 37 °C with 5% CO_2_.

### 4.4. Protein Extraction and Western Blot Analysis

Cells were lysed with 1 × Passive Lysis Buffer (1 × PLB; Promega) according to the manufacturer’s instructions. For Western blotting, membranes were blocked with 5% skim milk for 1 h at RT, then incubated with anti-Merlin (Cell Signalling #6995; diluted 1:1000), anti-β-actin (Sigma–Aldrich #A5441; diluted 1:10,000), or anti-Myc (Cell Signalling #2278; diluted 1:1000) for 1 h at RT or 12–16 h at 4 °C. After washing, membranes were incubated with HRP-conjugated secondary antibodies diluted at 1:2500 for 15 min, followed by incubation with Clarity™ Western ECL Substrate (Bio-Rad, Mississauga, ON, Canada) for 1 min. Images were processed on an Amersham Imager 600 series (GE Healthcare, Chicago, IL, USA). Western blot band intensity is quantified using ImageJ (NIH, Bethesda, MD, USA).

### 4.5. Lentivirus Production and Stable Cell Line Generation

Stable cell lines were generated by lentivirus-mediated transgene introduction into HEK293T and HCT-116 cells. Mer-Intra-BS was cloned into the lentiviral pTRIPZ vector, which confers puromycin resistance and contains a Tet-on promoter that allows Doxycycline (Dox)-inducible transgene expression. For lentivirus production, HEK293T cells were grown to 90–100% confluence on a 60 mm plate and then transfected with 1 µg Mer-Intra-BS/pTRIPZ, 0.75 µg psPAX (encoding lentiviral packaging components), and 0.25 µg PMD2G (encoding components for the viral envelope) using Polyjet Transient Transfection Reagent according to the manufacturer’s instructions (SignaGen, Frederick, MD, USA). NaButryrate was added to the culture media to a final concentration of 10 mM to increase lentivirus production 24 h after transfection. Cells were grown for another 24 h, and media containing lentivirus was harvested, passed through a 0.45 μm filter, and concentrated using a Lenti-X Concentrator (Clontech/Takara, San Jose, CA, USA. A total of 100 µL of concentrated virus was used to infect HEK293T and HCT116 cells cultured with 8 µg/mL polybrene. Two days after infection, the infected cells were selected for 2 µg/mL puromycin.

### 4.6. Luciferase Assays

For luciferase assays, cells were transfected with Mer-Intra-BS alone or with other plasmids using Polyjet Transfection Reagent (SignaGen) and lysed with 1 × PLB (Promega). For protein lysates or purified biosensor fusion proteins, the Nano-Glo Live Cell Assay System (Promega) was used to measure luciferase activity with furimazine as the luciferin substrate according to the manufacturer’s instructions, as previously described [55]. The luminescent activity was measured using a Turner Biosystems 20/20 Luminometer (Promega), or GloMax Navigator Microplate Luminometer (Promega). All data for luciferase assays are presented as luminescence relative to no biosensor control (RTC).

### 4.7. Bioluminescent Imaging (BLI) Analysis

Increasing numbers (0.5–5 × 10^5^ cells) of HEK293T or HCT116 cells stably expressing Mer-Intra-BS were seeded into each well of a 12-well plate. Bioluminescent signals were measured using Perkin Elmer IVIS Ilumina III (Waltham, MD, USA) after adding Nano-Glo Live Cell substrate (Promega).

### 4.8. Monitoring Cell Density Dependent Activation of Mer-Intra-BS

Triplicates of HEK293T-Mer-Intra-BS cells at increasing densities (1–8 × 10^4^) were plated into the individual wells of a 24-well plate. Mer-Intra-BS induction was achieved with Dox (1 μg/μL) treatment for 2 days. Subsequently, protein extraction, concentration measurement, and luciferase analysis were performed. The relative light units (RLU) per microgram of protein lysate were calculated for each sample. The experiment was replicated twice to ensure reproducibility. The mean and standard deviation (S.D.) of RLU/μg were determined for each cell density. A statistical analysis, specifically Student’s t-test, was conducted to compare the RLU/μg values between the 1 × 10^4^ cell density and other cell densities.

### 4.9. Purification of His-Tagged Mer-Intra-BS

LgBiT-Merlin-SmBiT was subcloned into a pET28b vector. The construct was transformed into BL21 DE3 competent bacteria. A single colony was inoculated into 25 mL of 2xYT medium containing ampicillin, incubated in an incubator at 37 °C, and shaken at 250 rpm overnight. The bacteria grown overnight were diluted to an OD_600_ of 0.2 in 250 mL of 2xYT medium and incubated at 37 °C until the OD_600_ was between 0.6 and 0.8. Protein expression was induced with 0.3 mM IPTG (isopropyl β-D-1-thiogalactopyranoside) overnight at 25 °C. Bacterial cells were lysed by sonication, bacterial lysates were centrifuged to collect soluble fractions, and His-tagged proteins were isolated from the supernatant via Ni-NTA affinity purification. Proteins were concentrated using an Amicon Ultra-4 Centrifugal Filter Unit (Millipore–Sigma) in a standard buffer (30 mM Tris-HCl, pH 7.5, 150 mM NaCl, 5 mM MgCl2, and 3 mM DTT). Concentrated proteins were analysed by SDS-PAGE and stored at −80 °C.

## 5. Conclusions

Our newly established Merlin intramolecular biosensor is a useful tool for studying the biochemical and biological function of tumour suppressor Merlin both in vitro and in vivo. It will significantly impact our understanding of Merlin’s roles in cancer biology and therapy.

## Figures and Tables

**Figure 1 ijms-25-01527-f001:**
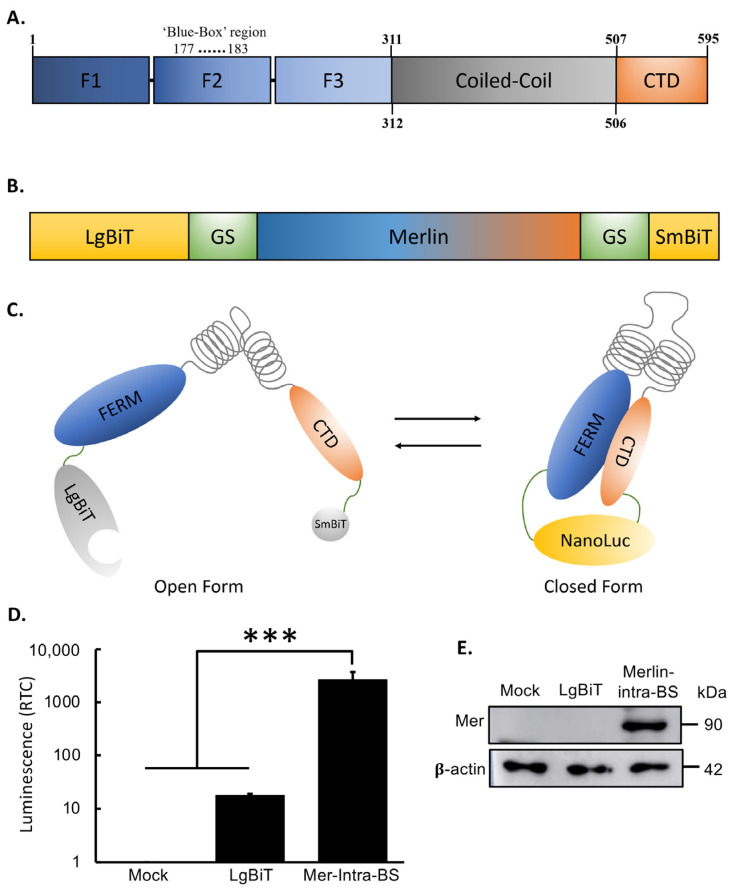
Establishment of the Mer-Intra-BS. (**A**) Domain structure of Merlin. The N-terminal FERM domain consists of three subdomains: F1, F2, and F3. The F2 subdomain contains a ‘blue-box’ region that is not found in ERM proteins but participates in binding with downstream Merlin effectors. (**B**) Construct design of the Mer-Intra-BS. The intramolecular Merlin-NanoBiT fusion protein was designed by fusing NanoBiT constituents LgBiT and SmBiT onto the N and C-terminus, respectively, of full-length Merlin. Each NanoBiT constituent is linked to Merlin through a flexible glycine/serine linker region. (**C**) Hypothesized mechanism of action for the Mer-Intra-BS. In the context of Mer-Intra-BS, luciferase complementation is expected when Merlin exists primarily in its closed conformation, but not when Merlin exists in an ‘open’ conformation. Therefore, the closed conformation of Mer-Intra-BS should display more luminescent activity. (**D**) Luciferase activity of the Mer-Intra-BS. The Mer-Intra-BS displays a dramatic (~150-fold, *p* < 0.0005) increase in luminescent activity compared to cells transfected with LgBiT alone, demonstrating that luciferase complementation occurs within the Mer-Intra-BS. Luminescence is presented, henceforth, as luminescence relative to control (Mock; RTC). Data are presented on a logarithmic scale. (**E**) The Mer-Intra-BS is detectable by Western blot with an anti-Merlin antibody. β-actin was used as an internal control. ***, *p* < 0.001, statistically significant.

**Figure 2 ijms-25-01527-f002:**
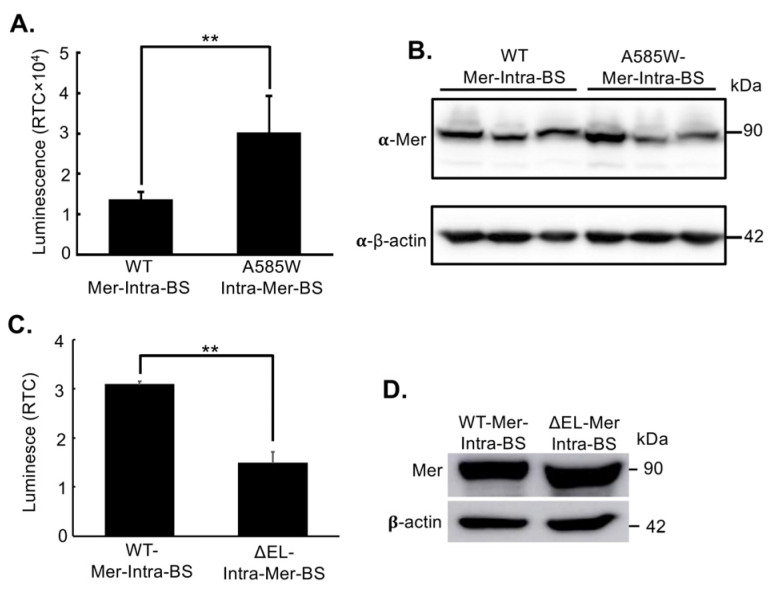
Validation of Mer-Intra-BS. (**A**) Validation of the Mer-Intra-BS using an A585W mutant biosensor. The A585W mutant promotes Merlin’s closed conformation by introducing a strong hydrophobic interaction between the N- and C-terminus of Merlin. Wild-type (WT) or A585W mutant biosensor plasmid was transfected into HEK293T cells, followed by luciferase assays. The protein lysates were subjected to Western blot analysis using an anti-Merlin antibody (**B**). Protein expression was detected by Western blot (**B**). (**C**) Validation of the Mer-Intra-BS using ΔEL-Intra-Mer-BS. The Mer-ΔEL-intra-BS promotes Merlin’s open conformation by disrupting the interaction between the N- and C-terminus of Merlin. No plasmid (mock) or plasmids expressing wild-type Mer-Intra-BS or ΔEL-Mer-Intra-BS were transfected into HEK293T cells, followed by luciferase assays. The protein lysates were subjected to Western blot analysis using an anti-Merlin antibody (**D**). RTC, Relative to control (mock), normalized by the levels of Mer-Intra-BS (lower panel) quantified by ImageJ. **, *p* < 0.01, statistically significant.

**Figure 3 ijms-25-01527-f003:**
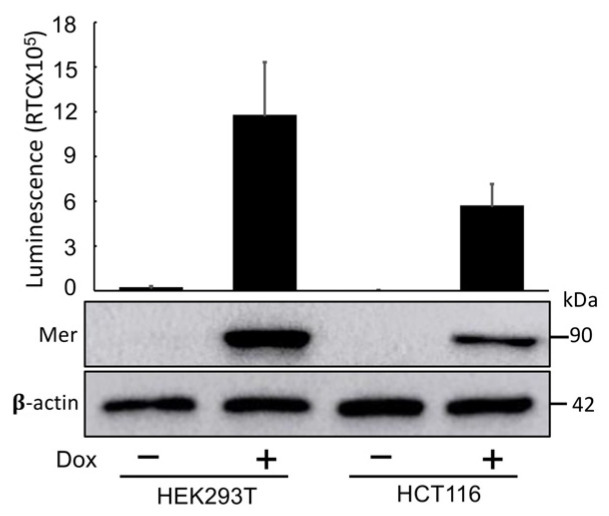
Cell lines with Dox-inducible expression of Mer-Intra-BS. Exposure to doxycycline dramatically increases expression (**upper panel**) and luminescent activity (**lower panel**) normalised by expression levels of Mer-Intra-BS in HEK293T and HCT16 cell lines stably transfected with Mer-Intra-BS under a Dox-inducible promoter. These stable cell lines are useful tools in studying Merlin conformation under various conditions.

**Figure 4 ijms-25-01527-f004:**
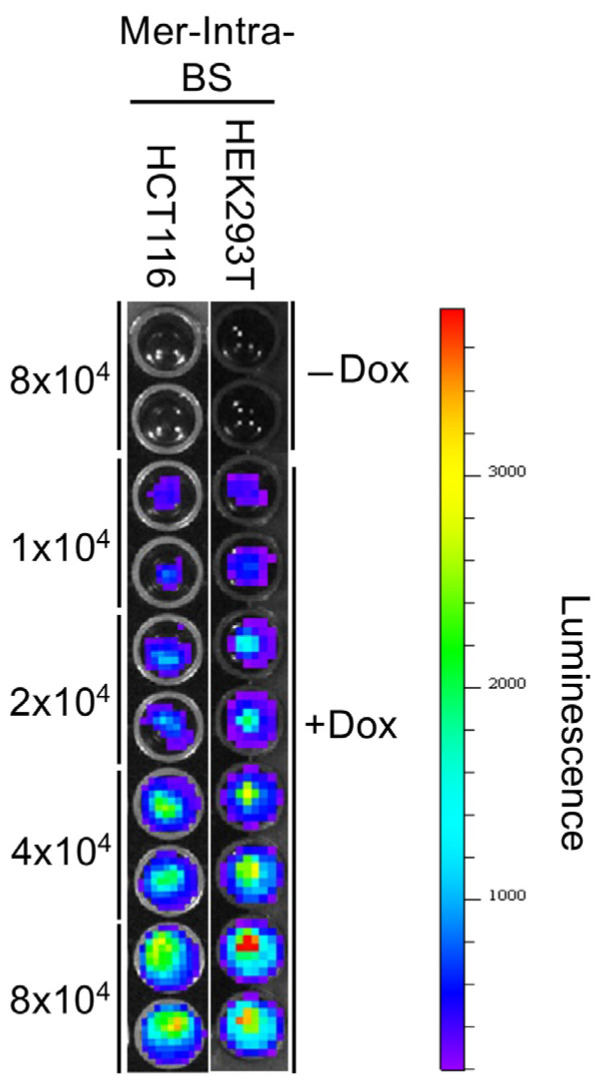
Bioluminescent imaging analysis of Mer-Intra-BS. Duplicates of an increasing number (1–8 × 10^4^) of HEK293T or HCT116 stably expressed Dox-inducible Mer-Intra-BS were seeded into a 24-well plate, followed by incubation in the presence of Dox (+Dox, 1 μg/mL) for 2 days. As a control, duplicates of 8 × 10^4^ cells were also seeded and incubated in the absence of Dox (—Dox) for 2 days. BLI was analyzed after adding a furimazine substrate. A heatmap of signal counts represents luminescence.

**Figure 5 ijms-25-01527-f005:**
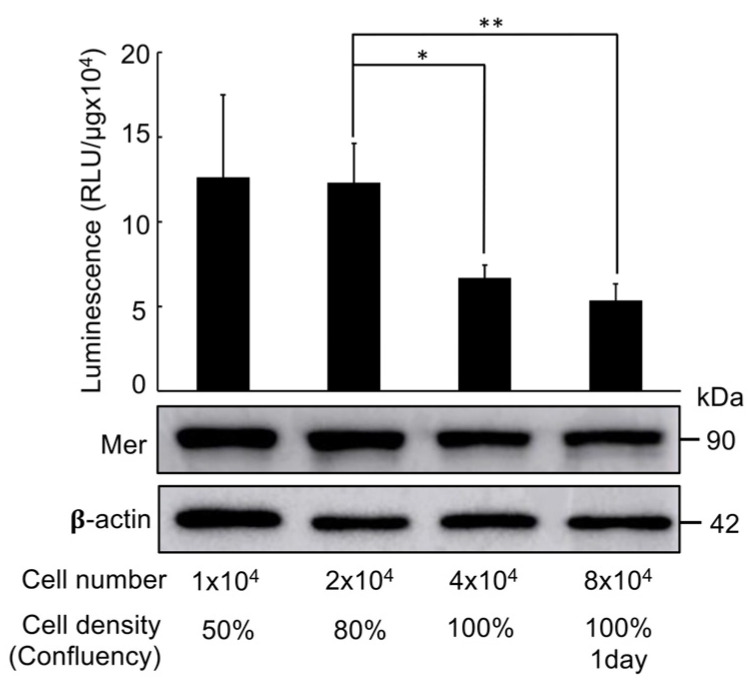
Density-dependent luciferase activity of the Mer-Intra-BS. Triplicates of HEK293T cells were stably transfected with Mer-Intra-BS under a Dox-inducible promoter and were seeded into a 24-well plate, followed by inducing Mer-Intra-BS by Dox (1 μg/mL) for 2 days. Following lysis, protein lysates were quantified, and luciferase activity was measured. (**Upper panel**): Luciferase assay. The protein concentration in protein lysates was measured. Relative luminescence (RLU) was normalised by protein concentration (RLU/μg) and ImageJ quantified Mer-Intra-BS (lower panel) levels. (**Lower Panel**): Western blot analysis of Mer-Intra-BS expression. We subjected 10 µg of lysates to Western blot analysis using an anti-Merlin antibody. Anti-β-actin was used as an internal loading control. *, *p* < 0.05; **, *p* < 0.01, statistically significant.

**Figure 6 ijms-25-01527-f006:**
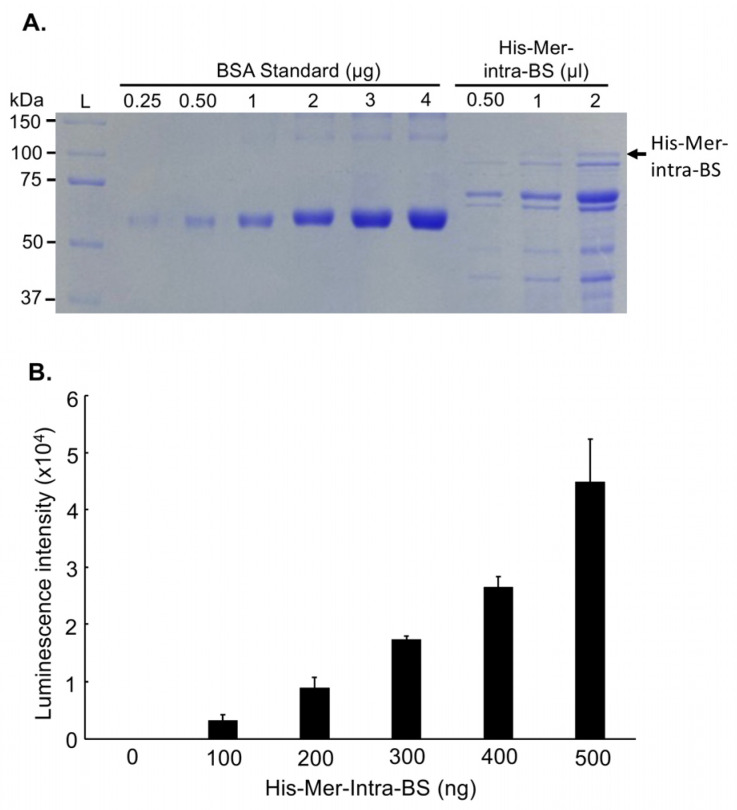
Characterization of Mer-Intra-BS in vitro. (**A**) Purification of Mer-Intra-BS. The Mer-Intra-BS was cloned into the PET28B(+) vector at NDE1 and BAMH1 restriction enzyme cutting sites to append a 6xHistidine tag. The His-Mer-Intra-BS was purified by affinity chromatography and quantified alongside BSA standards (0.25–4 μg). (**B**) Luciferase assays. Increasing amounts (0–500 ng) of purified His-Mer-Intra-BS were subjected to a luciferase assay in vitro. Relative luminescent units after subtracting the background (0 ng His-Mer-Intra-BS).

## Data Availability

No extra data except the data presented in the manuscript are presented.

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
