# Peer review of "Development of Novel Bioluminescent Biosensors Monitoring the Conformation and Activity of the Merlin Tumour Suppressor"

_ijms, 2024, doi:10.3390/ijms25031527_

Round 1

Reviewer 1 Report

Comments and Suggestions for Authors

In their paper entitled "Development of novel bioluminescent biosensors monitoring conformation and activity of Merlin tumor suppressor," Pipchuk et al., describe the design and validation of a split-luciferase biosensor system as a biosensor for Merlin, a key player in contact inhibition of cell proliferation (CIP) and frequently deactivated in several cancer types such as schwannomas, meningiomas and ependymomas. As such studying the protein, Merlin, is made difficult due to its diverse subcellular localization, lack of intrinsic catalytic activity, and frequent conformation changes. Specifically, the authors describe a Merlin biosensor that enables precise quantification of the conformation and activity of Merlin, both in vitro and in live cells.

Comments/suggestions:

1.     The abstract does not provide any information on how the authors designed the Merlin biosensor.

2.     Page 2, line # 54, authors may want to change “sequential” to “sequence”

3.     ** While authors have evaluated the intrinsic bioluminescence activity of LgBit alone and Mer-Intra-BS (and report ~150 fold increase in activity), however, it is may be useful to measure activity of construct containing LgBit fused with Merlin (but lacks the SmBit). It is possible that the presence of Merlin increases the intrinsic bioluminescence activity of LgBit alone.

4.     While authors have performed detailed validation of the Mer-Intra-BS, these could be improved through quantification of the protein expression western blots and normalization of bioluminescence values with protein expression values. This is particularly important since the increase (with the A585W mutant) and decrease (with the deltaEL mutant) in bioluminescence activity is only about two folds (Figure 2), respectively.

5.     Page 6, line 190-192: Authors write “Notably, this increase in luminescent activity was not associated with increased expression of the Mer-Intra-BS as measured by western blot (Figure 3, lower 191 panel),”. However, the western blot clearly shows a higher expression of Mer-Intra-BS in the case of PAK1-KD and PAK1-CA, compared to Mer-Intra-BS alone. The point #4 raised above could be applicable to data shown in Figure 3 as well, especially given the much higher Mer-Intra-BS expression levels shown in the bottom panel.

6.     Related to the point above (# 5), while using KD and CA PAK1 are informative, the authors may also include a non-mutant, wild type PAK1 in the assay. Also, since the authors claim that the impact of PAK1 expression is through the phosphorylation of residue S518, authors may want to include a phosphorylation deficient and a phosphomimetic mutant of Mer-Intra-BS to confirm this.

7.     Quantification of western blot data may also help resolve the difference in the bioluminescence activity of the Mer-Intra-BS in HEK293T and HCT116 cells under dox induction.

8.     Page 6, line 204: Authors write “Live cell bioluminescent imaging.”. However, this appears to be a standalone phrase without any description. Perhaps this is the subheading for the next section.

9.     Page 8, line # 232-234: Authors claim “Intriguingly, despite maintaining comparable expression levels at different cell densities, the signal experiences a significant reduction as cell density increases (2-8x104 cells, 80-100% confluency) (Figure 6).” However, above 3-fold decrease in bioluminescence could very well be attributed to a decrease in the Mer-Intra-BS expression levels as shown in the western blot image in the figure.

10.  Page 9, line # 253-254: Authors write “Notably, luciferase activity exhibited a positive correlation with the quantities of purified His-Mer-intra-BS used in the assays (Figure 7B).” This likely indicates that, at least a fraction of, Mer-Intra-BS is present in the ‘closed’ conformation without the need for phosphorylation of S518 or binding of a partner protein, as these are highly unlikely to occur in the recombinantly purified protein.

11.  Also, the authors may want to show background (observed with 0 ng of the biosensor) subtracted bioluminescence values in Figure 7.

12.  While authors claim a positive correlation between protein levels and bioluminescence activity, such analysis (i.e. correlation value) are not reported in either the text or in the figure.

13.  Overall, the key conclusion that the design biosensor reports both the conformation as well as activity of Merlin appears to be questionable in the absence of quantification of protein expression level.

Reviewer 2 Report

Comments and Suggestions for Authors

The authors generate a merline activity reporter that will be useful in studying contact inhibition, cell motility or general cytoskeletal studies. Some of the comments are below:

1. in some of the overexpression experiments, the levels are significantly different between groups. is it possible to normalize reporter activity with expression (e.g Figure 3, 4,6) ? in general, does the endogenous level affect the activity? Could they show with KD of merlin?

2. in figure 3: the authors can include the western levels for PAK. if possible, please include a better image for actin since it shows up as line and not very discernable as to wells/lanes.

3. Does merlin expression change with cell density? this goes to Figure 6. did they load same amount of protein? since the cell numbers were different how do they normalized this?

Round 2

Reviewer 1 Report

Comments and Suggestions for Authors

While authors write in their response letter that they have normalized the luciferase activity values (shown in various figures) with western blot-based protein expression values, I don't see any difference in the numbers/bar heights in the figures before and after revision. The authors may want to show how those values were affected after normalization with the protein expression levels.

Round 3

Reviewer 1 Report

Comments and Suggestions for Authors

Differences in the data/graphs due to the revision are now clear.